# MicroRNA-146b-5p Suppresses Pro-Inflammatory Mediator Synthesis via Targeting TRAF6, IRAK1, and RELA in Lipopolysaccharide-Stimulated Human Dental Pulp Cells

**DOI:** 10.3390/ijms24087433

**Published:** 2023-04-18

**Authors:** Peifeng Han, Keisuke Sunada-Nara, Nobuyuki Kawashima, Mayuko Fujii, Shihan Wang, Thoai Quoc Kieu, Ziniu Yu, Takashi Okiji

**Affiliations:** Department of Pulp Biology and Endodontics, Graduate School of Medical and Dental Sciences, Tokyo Medical and Dental University (TMDU), Tokyo 113-8549, Japan; hpfendo@tmd.ac.jp (P.H.); m.fujii.endo@tmd.ac.jp (M.F.); wang.endo@tmd.ac.jp (S.W.); kieuendo@tmd.ac.jp (T.Q.K.); yu.endo@tmd.ac.jp (Z.Y.); t.okiji.endo@tmd.ac.jp (T.O.)

**Keywords:** microRNA-146b-5p, human dental pulp cell, NF-κB signaling pathway, pro-inflammatory cytokine

## Abstract

MicroRNA-146b-5p (miR-146b-5p) is up-regulated during and to suppress the inflammation process, although mechanisms involved in the action of miR-146b-5p have not been fully elucidated. This study examined the anti-inflammation effects of miR-146b-5p in lipopolysaccharide (LPS)-stimulated human dental pulp cells (hDPCs). An increase in human miR-146b-5p (hsa-miR-146b-5p) expression following the mRNA expression of pro-inflammatory cytokines was observed in LPS-stimulated hDPCs. The expression of hsa-miR-146b-5p and pro-inflammatory cytokines was down-regulated by a nuclear factor-kappa B (NF-κB) inhibitor, and the expression of hsa-miR-146b-5p was also decreased by a JAK1/2 inhibitor. Enforced expression of hsa-miR-146b-5p abolished phosphorylation of NF-κB p65 and down-regulated the expression of pro-inflammatory cytokines and NF-κB signaling components, such as interleukin-1 receptor-associated kinase 1 (IRAK1), tumor necrosis factor receptor-associated factor 6 (TRAF6), and REL-associated protein involved in NF-κB (RELA). Expression of rat miR-146b-5p (rno-miR-146b-5p) and pro-inflammatory cytokine mRNA was also up-regulated in experimentally-induced rat pulpal inflammation in vivo, and rno-miR-146b-5p blocked the mRNA expression of pro-inflammatory mediators and NF-κB signaling components in LPS-stimulated ex vivo cultured rat incisor pulp tissues. These findings suggest that the synthesis of miR-146b-5p is controlled via an NF-κB/IL6/STAT3 signaling cascade, and in turn, miR-146b-5p down-regulates the expression of pro-inflammatory mediators by targeting TRAF6, IRAK1, and RELA in LPS-stimulated hDPCs.

## 1. Introduction

Dental pulp is a vascularized and innervated connective tissue localized inside the pulp chamber. It plays a crucial role in nourishment, repair, and defense against exogenous stimuli in the tooth [1]. Invasions from the oral environment such as bacteria or its metabolites from caries or trauma trigger defensive inflammatory responses [2]. The dentin/pulp complex is associated with these inflammatory responses, which involve numerous types of cells including odontoblasts, fibroblasts, endothelial cells, and immune cells [3]. Several pattern recognition receptors are involved in the response process, such as Toll-like receptors (TLRs) present in these cell types, which are responsible for detecting bacterial components and recognizing the pathogen and host-associated molecular patterns. TLR ligand binding causes the nuclear factor-kappa B (NF-κB) cascades to be activated [3,4], further inducing the synthesis of pro-inflammatory mediators such as cytokines and chemokines [5].

Pulp inflammation, like a coin, has two sides. On the one side, the increased generation of pro-inflammatory cytokines may cause irreversible tissue damage and lead to pulp necrosis [6]. On the other side, low-grade inflammation may boost regenerative mechanisms including angiogenic and stem cell processes [3,7]. Thus, controlling the synthesis of pro-inflammatory cytokines and maintaining a balance between pro- and anti-inflammatory mediators are of great importance to the inflammation process [8].

MicroRNAs (miRNAs) are single-stranded, endogenous, non-coding RNAs with a length of 20 to 25 nucleotides that act as gene regulators post-transcriptionally [9]. These tiny regulatory RNAs individually or corporately bind to one or more target messenger RNAs (mRNAs), thereby mediating translational repression or mRNA degradation [10,11,12]. miRNAs are anticipated to affect the activity of approximately 30% of all protein-coding genes and have been shown to act as important regulators in almost all cellular activities, including inflammation [11], immune response [12], and osteoclastic bone resorption [13]. miRNAs are reported to improve the efficacy of reprogramming potential of mouse and human somatic cells [14]. The generated induced pluripotent stem cells may possess immunomodulatory capacity [15], which will contribute to establishing new therapeutic strategies to regulate inflammation and recover the integrity of the dental pulp tissue [16]. Meanwhile, the synthesis of various miRNAs can be induced through the inflammation process; miRNAs such as microRNA-146b (miR-146b), miR-155, and miR-21 act to further modulate the expression of molecules that control the inflammatory response [17,18]. In particular, extensive research has shown the potential of miR-146b as an anti-inflammatory regulator in different cell types such as monocytes [19], airway smooth muscle cells [20], endothelial cells [21], and gingival fibroblasts [22].

Previous research evaluated the expression of miRNAs in dental pulp inflammation and reported that several miRNAs were up/down-regulated in inflamed dental pulp compared with those in normal pulp, which suggests that miRNAs play a multifaceted and distinctive role in pulpal inflammation and immunology [23]. We formerly reported 38 elevated miRNAs in lipopolysaccharide (LPS)-stimulated human dental pulp cells (hDPCs) via a miRNA array, from which human miR-146b-5p (hsa-miR-146b-5p) was notable for exhibiting up-regulation by more than 1.5-fold [17]. In the present study, we further evaluated miR-146b-5p with the hypothesis that miR-146b-5p is induced in LPS-stimulated hDPCs and works as a negative regulator through the progression of inflammation. We examined whether miR-146b-5p negatively modulates pro-inflammatory cytokine expression and revealed intracellular signaling mechanisms influenced by miR-146b-5p. These novel findings may offer a further understanding of the role that miRNAs have in dental pulp inflammation and potential therapeutic targets for the treatment of pulpitis.

## 2. Results

### 2.1. Expression of miR-146b-5p and mRNA of Pro-Inflammatory Mediators Was Up-Regulated in LPS-Stimulated hDPCs

Hsa-miR-146b-5p expression was significantly up-regulated at 12 h after LPS stimulation and decreased thereafter in hDPCs (*p* < 0.05, Figure 1A). mRNA expression of pro-inflammatory cytokines and chemokines including interleukin 6 (IL6), IL8, monocyte chemotactic protein 1 (MCP1), IL1α (gene symbol *IL1A*), IL1β (gene symbol *IL1B*), and tumor necrosis factor alpha (TNFα; gene symbol *TNFA*) peaked at 2–6 h after LPS stimulation (*p* < 0.05 or 0.01, Figure 1B), which was earlier than the peak of hsa-miR-146b-5p expression.

### 2.2. NF-κB and JAK/STAT3 Signaling Were Involved in the Expression of Pro-Inflammatory Cytokines and miR-146b-5p in LPS-Stimulated hDPCs, Respectively

mRNA expression of pro-inflammatory cytokines *IL6*, *IL8*, *MCP1*, *IL1A*, and *IL1B* as well as the protein expression of IL6, IL8, and MCP1 was induced by 6 h of LPS stimulation, and was significantly down-regulated by BAY 11-7085, a typical inhibitor of NF-κB (*p* < 0.05 to 0.0001, Figure 2A,B). The increase in hsa-miR-146b-5p expression induced by 6 h of LPS stimulation was also abolished not only by BAY 11-7085 but also by ruxolitinib, a representative JAK1/2 inhibitor (*p* < 0.05 or 0.01, Figure 2C).

### 2.3. miR-146b-5p Down-Regulated NF-κB Signaling and Synthesis of Pro-Inflammatory Cytokines in LPS-Stimulated hDPCs

Increased expression of hsa-miR-146b-5p in hDPCs transfected with mirVana miRNA mimic for hsa-miR-146b-5p (miR-146b mimic) was confirmed (*p* < 0.01, Figure 3A). Promoted NF-κB activity and expression of phosphorylated NF-κB p65 by LPS stimulation were significantly down-regulated by over-expression of miR-146b mimic (*p* < 0.05 or 0.01, Figure 3B,C), as well as phosphorylated IKKα/β and phosphorylated IκBα (Appendix A). Furthermore, up-regulated mRNA expression of *IL6*, *IL8*, *MCP1*, *IL1A*, *IL1B*, and *TNFA*, as well as protein expression of IL6, IL8, and MCP1, was significantly depressed by over-expression of hsa-miR-146b-5p in LPS-stimulated hDPCs (*p* < 0.05 to 0.0001, Figure 3D,E).

### 2.4. miR-146b-5p Targeted TRAF6, IRAK1, and RELA

Tumor necrosis factor receptor-associated factor 6 (TRAF6), interleukin-1 receptor-associated kinase 1 (IRAK1), and REL-associated protein (RELA/NF-κB p65) are essential mediators of NF-κB signaling and are the predicted target genes of miR-146b-5p (TargetScanHuman). Over-expression of hsa-miR-146b-5p significantly down-regulated mRNA and protein expression of TRAF6, IRAK1, and RELA in LPS-stimulated hDPCs (*p* < 0.05 to 0.001, Figure 4A,B,D,E,G,H). Similar down-regulated expression of TRAF6, IRAK1, and RELA in LPS-stimulated hDPCs was confirmed by immunocytological evaluation (Figure 4C,F,I). In addition, mRNA and protein expression of TRAF6, IRAK1, and RELA was independent of the presence of LPS (*p* > 0.05, Figure 4A,B,D,E,G,H and Appendix A). Furthermore, over-expression of hsa-miR-146b-5p significantly down-regulated the luciferase activity in hDPCs transfected with a reporter vector containing hsa-miR-146b-5p target sequence in the RELA 3′-UTR (*p* < 0.05, Figure 4J,K), whereas such phenomena were not observed in hDPCs transfected with a reporter vector containing mutated hsa-miR-146b-5p target region in the RELA 3′-UTR (*p* > 0.05, Figure 4J,K).

### 2.5. miR-146b-5p Down-Regulated Pro-Inflammatory Cytokines in Rat Dental Pulp Tissue via Targeting TRAF6, IRAK1, and RELA Ex Vivo

Pulpal inflammation was experimentally induced in the rat incisor pulp by application of LPS (Figure 5A). Expression of rat miR-146b-5p (rno-miR-146b-5p), as well as mRNA expression of pro-inflammatory cytokines *Il6*, *Mcp1*, *Il1a*, *Il1b*, and *Tnfa*, was significantly elevated in LPS-applied rat incisor pulp tissue at 3 to 6 h after LPS stimulation (*p* < 0.05 to 0.0001, Figure 5B,C). Furthermore, significant up-regulation of rno-miR-146b-5p via transfection with mirVana miRNA mimic for rno-miR-146b-5p (miR-146b mimic) was observed in LPS-stimulated ex vivo cultured rat incisor pulp tissue (*p* < 0.0001, Figure 6A). Over-expression of rno-miR-146b-5p induced significant down-regulation of phosphorylated NF-κB p65 expression (*p* < 0.01, Figure 6B) and mRNA expression of pro-inflammatory cytokines including *Il6*, *Mcp1*, *Il1a*, and *Il1b* (*p* < 0.05 to 0.0001, Figure 6C). mRNA and protein expression of Traf6, Irak1, and Rela was also significantly diminished by over-expression of rno-miR-146b-5p (*p* < 0.05 or 0.01, Figure 6D–L).

## 3. Discussion

In the present study, we revealed that in hDPCs, hsa-miR-146b-5p expression was significantly elevated at 12 h after LPS stimulation and decreased thereafter (Figure 1A). The augmentation of hsa-miR-146b-5p under LPS exposure is consistent with existing research showing that miR-146b-5p synthesis was induced in LPS-stimulated THP-1 cells [19], leukocytes [24], and monocytes [25]. In LPS-stimulated hDPCs, the increase of hsa-miR-146b-5p expression appeared to occur following the up-regulation of pro-inflammatory cytokines including *IL6*, *IL8*, *MCP1*, *IL1A*, *IL1B*, and *TNFA* (Figure 1B), which could occur via the NF-κB signaling pathway. Consistently, in vivo expression of rno-miR-146b-5p and pro-inflammatory cytokines was also provoked in the LPS-induced rat pulp inflammation model (Figure 5B,C). These results suggest that miR-146b-5p was synthesized upon LPS stimulation in hDPCs and rat dental pulp tissue and that miR-146b-5p plays a vital role in the regulation of pulpal inflammatory reactions, which were mediated by various pro-inflammatory cytokines.

TLR4 is regarded as the most significant TLR for initiating robust immunological responses in response to LPS exposure [26], and the binding of LPS to TLR4 induces activation of the NF-κB signaling pathway, resulting in the generation of pro-inflammatory cytokines [27]. It has been reported that TLR4 is present in various cells such as hDPCs [28] and its expression could be up-regulated by LPS stimulation [17,29]. Our results showed that in LPS-stimulated hDPCs, both mRNA and protein expression of pro-inflammatory cytokines was significantly increased, while it was drastically decreased with the application of NF-κB inhibitor BAY 11-7085 (Figure 2A,B). These results indicate that the NF-κB signaling pathway was activated and involved in the synthesis of pro-inflammation cytokines in LPS-stimulated hDPCs. Moreover, the application of BAY 11-7085, as well as ruxolitinib, considerably reduced hsa-miR-146b-5p expression (Figure 2C), suggesting that the NF-κB signaling pathway, along with the JAK/STAT3 signaling pathway, contributes to the process of miR-146b-5p synthesis. IL6 is a major inflammatory cytokine synthesized through the activation of NF-κB [30] and CCAAT enhancer-binding protein beta [31], which is also an upstream initiator of JAK/STAT3 signaling pathway [32,33,34]. During the dynamic expression process of miRNA and inflammatory cytokines in hDPCs, mRNA expression of IL6 showed an earlier peak compared with the expression of hsa-miR-146b-5p (Figure 1A,B) which indicates that synthesis of IL6 might contribute to promoting the production of hsa-miR-146b-5p. The hypothesis is supported by previous reports that IL6 activates STAT3 while two putative STAT3 binding sites have been reported in the miR-146b promoter region [35], and the binding of STAT3 to the miR-146b promoter induces transcription of pre-miR-146b [25]. Taking these findings together, LPS activates NF-κB signaling while induced IL6 expression stimulates STAT3 activation, which suggests that hsa-miR-146b-5p synthesis in LPS-stimulated hDPCs might be regulated by the NF-κB and JAK/STAT3 signaling pathways.

NF-κB subunit phosphorylation plays a crucial role in the activation and function of NF-κB signaling [36]. The phosphorylation events that regulate gene transcription activity are caused by signals from components of the NF-κB signaling pathway or factors from outside the pathway, and the phosphorylation events contribute to the selective regulation of NF-κB transcriptional activity [36]. p65 is one of the important subunits of NF-κB and its phosphorylation induces a conformational change, which influences the ubiquitination and stability of p65, as well as interactions between proteins [37]. In the present study, over-expression of miR-146b-5p down-regulated NF-κB reporter activity in hDPCs (Figure 3B) as well as expression of NF-κB p65 phosphorylation promoted by LPS stimulation in both hDPCs and ex vivo cultured rat incisor pulp tissue (Figure 3C and Figure 6B), which indicates that miR-146b-5p may be a negative regulator of the NF-κB signaling pathway. We further revealed that over-expression of miR-146b-5p in hDPCs and ex vivo cultured rat incisor pulp tissue induced the decrease of mRNA and protein expression of LPS-dependent pro-inflammatory cytokines (Figure 3D,E and Figure 6C). These findings suggest that miR-146b-5p plays a critical role in the negative regulation of pro-inflammatory cytokine expression by interfering with the NF-κB signaling pathway, and may exert anti-inflammatory effects in dental pulp inflammation to avoid excessive tissue damage.

By targeting signal transduction proteins involved in intracellular signaling after pathogen recognition, miRNAs could have a major impact on the subsequent inflammatory response by specifically targeting mRNAs that encode certain pro-inflammatory cytokines [18]. It has been reported that miR-146b targets are abundant in TLR/NF-κB signaling [25], in which LPS/TLR4 signaling recruits TRAF6 and IRAK1 complex [38], then further promotes NF-κB activation, leading to the generation of pro-inflammatory cytokines. TRAF6 works as a signal transducer in the NF-κB pathway that activates IκB kinase [39] and interacts with IRAK1, which is phosphorylated by IRAK4 [40,41]. TRAF6 and IRAK1 have been reported to be direct targets of miR-146b in TLR/NF-κB signaling (TLR4 and MyD88 have also been reported to be the direct targets of miR-146b-5p [25]). Our results showed that over-expression of hsa-miR-146b-5p down-regulated mRNA and protein expression of TRAF6 and IRAK1 in LPS-stimulated hDPCs (Figure 4A–F). In addition to the reported targets of miR-146b, we here report for the first time that mRNA and protein expression of RELA was also down-regulated via over-expression of hsa-miR-146b-5p (Figure 4G–I). Our results indicated the direct binding sites of hsa-miR-146b-5p in the RELA 3′-UTR sequence (Figure 4J). The luciferase activity in hDPCs transfected with a reporter vector containing the hsa-miR-146b-5p target sequence in the RELA 3′-UTR was considerably reduced by over-expression of hsa-miR-146b-5p (Figure 4K), while in the hDPCs transfected with a reporter vector containing mutated hsa-miR-146b-5p target region in the RELA 3′-UTR, the down-regulation of luciferase activity was abolished (Figure 4K). These findings suggest that RELA is a direct target of hsa-miR-146b-5p.

Similar to our observations in LPS-stimulated hDPCs, over-expression of rno-miR-146b-5p corresponded with down-regulated mRNA and protein expression of TRAF6, IRAK1, and RELA in LPS-stimulated ex vivo cultured rat incisor pulp tissue (Figure 6D–L). Findings from hDPCs and ex vivo cultured rat incisor pulp tissue identified miR-146b-5p as an anti-inflammatory miRNA that is capable of reducing inflammatory signals through multiple targeting mechanisms directed at the receptor and its adaptor proteins. Our results are consistent with the perspective that rather than operating repression of a single target, miRNAs could carry out their significant functions by regulating many targets implicated in a common signaling cascade [42,43]. Ex vivo results also indicated the potential for the application of miR-146b-5p in the clinical treatment of pulp inflammation, although further study and development are required.

In the larger context of oral health, our study provides important insights into the prevalence and impact of pulp inflammation and offers new possibilities for the development of novel therapies that target the underlying molecular mechanisms. By advancing our understanding of the complex mechanisms involved in inflammation, our study has the potential to the possibility of miR-146b-5p being therapeutically induced or applied to inflamed pulp tissue to aid the development of anti-inflammatory treatments. However, limitation exists in lacking in vivo validation of the observed effects of miR-146b-5p in human pulpal inflammation. While the results obtained from in vitro experiments on hDPCs and ex vivo experiments on rat dental pulp tissue are promising, further studies are needed to confirm the efficacy and safety of miR-146b-5p as a potential therapeutic agent in vivo. Additionally, future studies may also explore the role and targets of miR-146b-5p in different cells and its feasibility of delivering to inflamed pulp tissue in a controlled and targeted manner.

## 4. Materials and Methods

### 4.1. Cell Culture

All methods and experiments were approved by the Ethical Committee of Tokyo Medical and Dental University and carried out in accordance with the relevant Ethical Guidelines and Regulations for Clinical Studies (Reference number D2014-039). All participants provided informed consent in accordance with the Ethical Guidelines and Regulations for Clinical Studies. hDPCs were isolated from dental pulp tissue of human third molars and cultured in α minimum essential medium (αΜΕΜ; FUJIFILM Wako Pure Chemical, Osaka, Japan) containing 10% fetal bovine serum (FBS; Thermo Fisher Scientific, Waltham, MA, USA) and 1% penicillin/streptomycin (FUJIFILM Wako Pure Chemical) under a standard condition (37 °C, 5% CO_2_). The culture medium was changed at 3-day intervals, and cells from passages 2–7 were used. To provide an inflammatory stimulus to hDPCs, LPS (*Escherichia coli* O111:B4, 100 ng/mL; Sigma-Aldrich, St. Louis, MO, USA) was applied. BAY 11-7085 (CAS Number: 196309-76-9, 1 μM; Cayman Chemical, Ann Arbor, MI, USA), a specific and potent NF-κB signaling inhibitor [44,45], was used to block the NF-κB signaling pathway. Ruxolitinib (CAS Number: 941678-49-5, 10 μM; Cayman Chemical), a specific and potent JAK1/2 inhibitor [46,47], was used to block the JAK/STAT3 signaling pathway.

### 4.2. miR-146b-5p Mimic Transfection

The mirVana miRNA mimic for hsa-miR-146b-5p (miR-146b mimic; Thermo Fisher Scientific) was transfected into hDPCs via Lipofectamine RNAiMAX transfection reagent (Thermo Fisher Scientific). The mirVana miRNA mimic Negative Control #1 (miR-146b NC; Thermo Fisher Scientific) was used as a control.

### 4.3. LPS-Induced Rat Pulp Inflammation In Vivo

All animal experiments were conducted under the authorized parameters of the Institutional Committees for Animal Experiments at Tokyo Medical and Dental University (TMDU) and all experimental protocols were approved by the Institutional Animal Care and Use Committee of TMDU (Reference number A2019-297C). All animal experiments were reported in compliance with the ARRIVE guidelines. Sprague Dawley rats (6 weeks, male, n = 9; Clea Japan, Tokyo, Japan) were housed under standard conditions (22 °C, 55% relative humidity, artificial illumination) and offered lab rat fodder and free access to water. Ketamine hydrochloride (50 mg/kg, intraperitoneally) and xylazine (20 mg/kg, intraperitoneally) were used for anesthesia. To generate experimental rat pulpitis, pulp tissue of the upper and lower incisors was exposed. LPS solution (1 μL, 10 mg/mL, dissolved in sterile saline) was applied to the pulp by sterile paper points. A light-curing resin (G-FIX; GC, Tokyo, Japan) was used to seal the cavities. For quantification of mRNA and miRNA expression, rats were sacrificed under CO_2_ at 0, 3, 6, 12, and 24 h, and pulp tissue was extracted and promptly preserved in RNAlater (Thermo Fisher Scientific) to stabilize RNA. For histological evaluation, rats were sacrificed under CO_2_ at 0 and 12 h. Upper and lower incisors were collected and fixed with 4% paraformaldehyde in phosphate-buffered saline (PBS) at 4 °C for 24 h and demineralized with EDTA for 14 days. The specimens were embedded with an embedding medium (OCT Compound; Sakura Finetek, Tokyo, Japan) and sliced into 10 μm frozen sections using a cryostat (CM3050; Leica Microsystems, Wetzlar, Germany); sections were used for hematoxylin and eosin (H&E) staining.

### 4.4. Ex Vivo miR-146b-5p Mimic Transfection

Sprague Dawley rats (6 weeks, male, n = 9; Clea Japan) were sacrificed under CO_2_ and pulp tissue from upper and lower incisors was immediately extracted and cultured in αMEM. The mirVana miRNA mimic for rno-miR-146b-5p (miR-146b mimic, Thermo Fisher Scientific) was transfected into rat incisor pulp tissue via Lipofectamine RNAiMAX transfection reagent (Thermo Fisher Scientific). The mirVana miRNA mimic Negative Control #1 (miR-146b NC; Thermo Fisher Scientific) was used as a control. Twenty-four h after transfection, rat incisor pulp tissues were stimulated with LPS (100 ng/mL) for 6 h. The specimens fixed with 4% paraformaldehyde in PBS at 4 °C for 24 h were embedded and sliced into 10-μm frozen sections as described above for further immunofluorescence analysis.

### 4.5. RT-qPCR

Total RNA in the hDPCs and rat incisor pulp tissues was extracted using QuickGene RNA cultured cell kit S (FUJIFILM Wako Pure Chemical) and TRIzol reagent (Thermo Fisher Scientific), respectively. cDNA was synthesized using PrimeScript™ RT Master Mix (Takara Bio, Kusatsu, Japan), and PCR products were amplified using synthesized cDNA, specific primers (Table 1), and GoTaq qPCR Master Mix (Promega, Madison, WI, USA) via the CFX96 Real-Time qPCR System (Bio-Rad, Kidlington, UK). Actin beta (*ACTB*) was used as an internal control. To evaluate miRNA, mirVana miRNA Isolation Kit (Thermo Fisher Scientific) was used to extract total RNA. cDNA was synthesized using TaqMan microRNA Assays (Thermo Fisher Scientific) and a microRNA Reverse Transcription Kit (Thermo Fisher Scientific) with specific RT primers for hsa-miR-146b-5p, rno-miR-146b-5p, and U6. Real-time qPCR was performed with TaqMan Universal Master Mix II, via a CFX96 Real-Time qPCR System (Bio-Rad). U6 spliceosome RNA was used as an internal control.

### 4.6. Western Blotting

Cells and tissues were lysed by radioimmunoprecipitation buffer containing a phosphatase inhibition cocktail (PhosSTOP; Sigma-Aldrich) and a protease inhibition cocktail (cOmplete; Sigma-Aldrich). Sample lysates were electrophoresed on polyacrylamide gels containing sodium dodecyl sulfate (e-PAGEL; ATTO, Tokyo, Japan), and the gels were transferred to polyvinylidene difluoride membranes (Immobilon-P; Merck Millipore, Burlington, MA, USA) using a semidry transfer system (0.15 mA, 1 h, WSE-4040; ATTO). The blots were incubated with the following primary antibodies: anti-p65/RELA (1:1000, D14E12, monoclonal, rabbit; Cell Signaling Technology, Danvers, MA, USA), anti-phospho-NF-κB p65 (1:1000, Ser536, 93H1, monoclonal, rabbit; Cell Signaling Technology), anti-IKKα (1:1000, 3G12, monoclonal, mouse; Cell Signaling Technology), anti-IKKβ (1:1000, D30C6, monoclonal, rabbit; Cell Signaling Technology), anti-phospho-IKKα/β (1:1000, Ser176/180, 16A6 monoclonal, rabbit; Cell Signaling Technology), anti-IκBα (1:1000, L35A5, monoclonal, mouse; Cell Signaling Technology), anti-phospho-IκBα (1:1000, Ser32, 14D4, monoclonal, rabbit; Cell Signaling Technology), anti-TRAF6 (1:1000, GTX113029, polyclonal, rabbit; GeneTex, Irvine, CA, USA), anti-IRAK1 (1:500, GTX31253, polyclonal, rabbit; GeneTex), horseradish peroxidase (HRP)-conjugated anti-GAPDH (1:4000, PM053-7; Medical & Biological Laboratories, Nagoya, Japan), and HRP-conjugated anti-tubulin (1:4000, PM054-7; Medical & Biological Laboratories, Nagoya, Japan). HRP-conjugated anti-rabbit IgG (1:5000, W4011; Promega) and HRP-conjugated anti-mouse IgG (1:5000, W4021; Promega) were used as secondary antibodies. A chemiluminescent HRP substrate (Immobilon, Millipore) was used as a substrate for HRP. Images were obtained by a LAS-3000 mini-imaging system (Fujifilm, Tokyo, Japan). The pixel integrated density of the bands was quantified using ImageJ software v2 (National Institutes of Health, Bethesda, MD, USA) and the ratios of the density were calculated.

### 4.7. Cytometric Bead Array (CBA)

Cell culture medium supernatant was collected after conditional treatment. According to the manufacturer’s protocol for cytokine measurement, all samples were subjected to the BD CBA Human Flex Set (BD Biosciences, Franklin Lakes, NJ, USA). Capture beads coated with antibodies against IL6, IL8, and MCP1 were incubated in each sample. Phycoerythrin-conjugated antibodies were added to detect cytokines, and cytokine quantification was calculated on the basis of a standard curve plot. Fluorescence from phycoerythrin-conjugated antibodies was detected using the FACS Canto II flow cytometer (BD Biosciences) following the manufacturer’s instructions. Data were analyzed via FCAP Array Software v3.0 (BD Biosciences).

### 4.8. Luciferase Assay with NF-κB and RELA 3′-UTR Reporter Vectors

For the luciferase assay with NF-κB, pGL4.32 (luc2P/NF-κB-RE/Hygro) vector containing five copies of NF-κB response element (Promega) was used. The hDPCs were transfected with the reporter vector and miR-146b mimic or miR-146b NC using Lipofectamine 3000 transfection reagent (Thermo Fisher Scientific) and then stimulated with LPS (100 ng/mL). For the luciferase assay with RELA 3′-UTR reporter vector, synthesized RELA 3′-UTR containing wild-type or mutated hsa-miR-146b-5p target sequences (400 bps each; Eurofins Genomics, Ebersberg, Germany) were inserted into XhoI and HindIII sites of a pMIR-REPORT vector (Thermo Fisher Scientific). The reporter vectors along with miR-146b mimic or miR-146b NC were transfected into hDPCs via Lipofectamine 3000 transfection reagent (Thermo Fisher Scientific). Transfected cells were lysed with a luciferase cell culture lysis reagent and the luciferase activity was measured using luciferase assay system (Promega) and a luminometer (Luminescence PSN; ATTO).

### 4.9. Immunofluorescence

hDPCs were rinsed with PBS and fixed with 4% paraformaldehyde in PBS at 4 °C for 10 min. Fixed cells and sections were permeabilized with PBS containing 0.1% Triton X-100 and then blocked with 10% normal donkey serum for 30 min. The samples were incubated overnight with primary antibodies: anti-TRAF6 (1:500, GTX113029, polyclonal, rabbit; GeneTex), anti-IRAK1 (1:500, GTX31253, polyclonal, rabbit; GeneTex), and anti-RELA (1:1000, D14E12, monoclonal, rabbit; Cell Signaling Technology). Alexa Fluor 488-conjugated anti-rabbit IgG (1:500, donkey; Abcam, Cambridge, UK) was applied for visualization. Coverslips were mounted with a fluorescent mounting medium containing 4′,6-diamidino-2-phenylindole (DAPI) nuclear stain (Flouroshield Mounting Medium with DAPI; Abcam). A confocal laser scanning microscope (Leica TCS-SP8; Leica Microsystems) and LAS AF confocal software (Version 1.8.3; Leica Microsystems) were used to perform histological assessment.

### 4.10. Statistical Analysis

Statistical analysis was conducted with SPSS Statistics 28.0 (IBM, Armonk, NY, USA) and GraphPad Prism 8 (GraphPad Software, San Diego, CA, USA). Data normality and equality of variance were checked by the Shapiro–Wilk test and Levene’s test, respectively. Unpaired Student’s *t*-test, or Welch’s *t*-test was used for statistical analysis between two groups. Ordinary one-way analysis of variance (ANOVA) followed by Tukey’s post hoc test, Brown–Forsythe/Welch one-way ANOVA test followed by Dunnett’s T3 post hoc test, or Kruskal–Wallis test followed by Dunn’s post hoc test was used for statistical analysis among three groups or more. Values of *p* < 0.05 were considered statistically significant.

## 5. Conclusions

In conclusion, we here report that in LPS-stimulated hDPCs/rat dental pulp tissue, pro-inflammatory mediators were generated via activation of the NF-κB signaling pathway, while miR-146b-5p expression was controlled by the NF-κB and JAK/STAT3 signaling pathways. miR-146b-5p in turn negatively influenced the progression of inflammation via down-regulation of TRAF6, IRAK1, and RELA expression (Figure 7). Our findings point to the beneficial impacts of miR-146b-5p in the pulp inflammation process through the negative regulation of excessive inflammatory responses and offer a deeper understanding of how miR-146b-5p interacts with a rising number of negative mediators, especially during complex inflammation progression involving multiple receptor-signaling pathways.

## Figures and Tables

**Figure 1 ijms-24-07433-f001:**
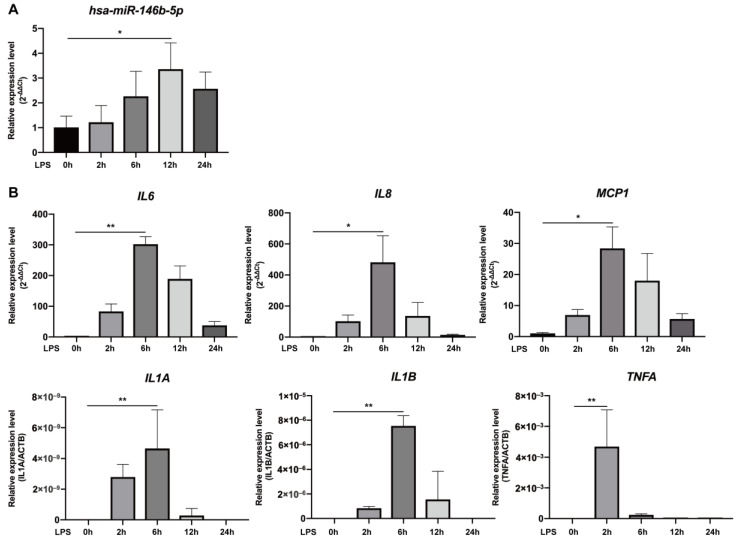
Expression of hsa-miR-146b-5p and pro-inflammatory mediator mRNAs is up-regulated in LPS-stimulated hDPCs. (**A**) Expression of hsa-miR-146b-5p was induced in LPS-stimulated hDPCs and significantly up-regulated at 12 h after LPS stimulation (mean ± SD, n = 4). (**B**) mRNA expression of pro-inflammatory mediators was induced in LPS-stimulated hDPCs. mRNA expression of *IL6*, *IL8*, *MCP1*, *IL1A*, and *IL1B* peaked at 6 h after LPS application. mRNA expression of *TNFA* peaked at 2 h after LPS application (mean ± SD, n = 4). * *p* < 0.05 and ** *p* < 0.01. LPS: lipopolysaccharide; hDPCs: human dental pulp cells.

**Figure 2 ijms-24-07433-f002:**
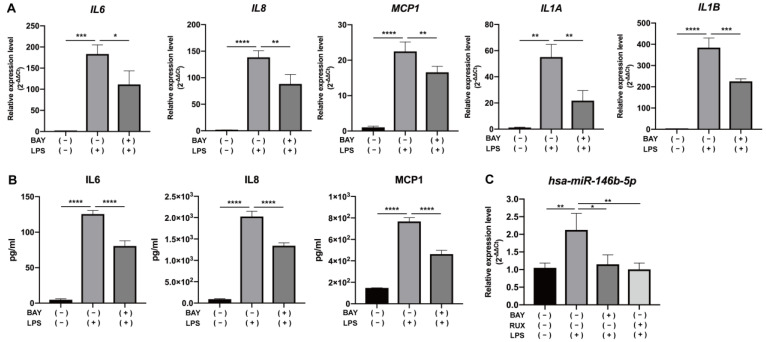
NF-κB signaling is involved in the expression of hsa-miR-146b-5p and pro-inflammatory mediator mRNAs in LPS-stimulated hDPCs, and JAK/STAT3 signaling is also involved in the expression of hsa-miR-146b-5p. (**A**) mRNA expression of *IL6*, *IL8*, *MCP1*, *IL1A*, and *IL1B* was up-regulated by LPS stimulation at 6 h and was abolished by NF-κB inhibitor BAY 11-7085 in hDPCs (mean ± SD, n = 4). (**B**) Protein expression of IL6, IL8, and MCP1 was up-regulated by LPS stimulation at 6 h and was down-regulated by BAY 11-7085 in hDPCs (mean ± SD, n = 4). (**C**) Expression of hsa-miR-146b-5p was up-regulated by LPS stimulation at 6 h and was down-regulated by BAY 11-7085 or JAK1/2 inhibitor ruxolitinib in hDPCs (mean ± SD, n = 3). * *p* < 0.05, ** *p* < 0.01, *** *p*  <  0.001, and **** *p* < 0.0001. LPS: lipopolysaccharide; hDPCs: human dental pulp cells; BAY: BAY 11-7085; RUX: ruxolitinib.

**Figure 3 ijms-24-07433-f003:**
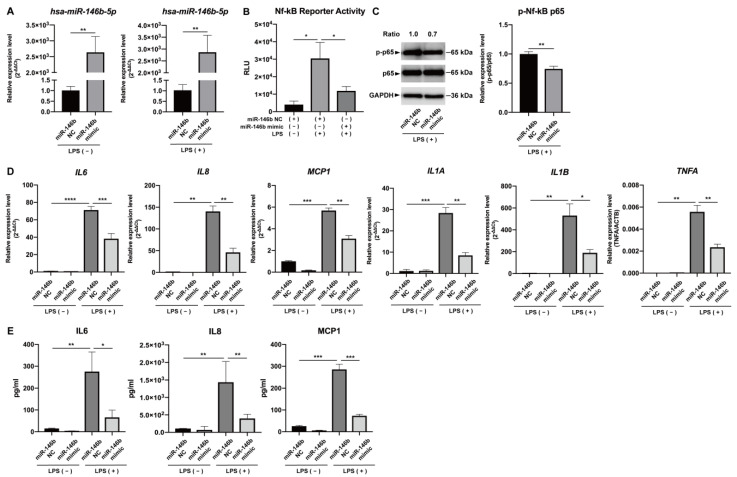
hsa-miR-146b-5p mimic down-regulates the activated NF-κB signaling and up-regulates expression of proinflammatory cytokines in LPS-stimulated hDPCs. (**A**) Transfection of hsa-miR-146b-5p mimic induced up-regulation of hsa-miR-146b-5p in hDPCs with or without LPS stimulation (mean ± SD, n = 4). (**B**) hsa-miR-146b-5p mimic significantly down-regulated the promoted NF-κB reporter activity in hDPCs under 2 h LPS stimulation (mean ± SD, n = 4). (**C**) hsa-miR-146b-5p mimic significantly down-regulated the promoted phosphorylation of NF-κB p65 expression in hDPCs under 2 h LPS stimulation (mean ± SD, n = 4). (**D**) hsa-miR-146b-5p mimic significantly down-regulated the promoted mRNA expression of *IL6*, *IL8*, *MCP1*, *IL1A*, *IL1B*, and *TNFA* in hDPCs under 2 h LPS stimulation (mean ± SD, n = 4). (**E**) hsa-miR-146b-5p mimic significantly down-regulated the promoted protein expression of IL6, IL8, and MCP1 in hDPCs under 2 h LPS stimulation (mean ± SD, n = 3). * *p* < 0.05, ** *p* < 0.01, *** *p* < 0.001, and **** *p* < 0.0001. LPS: lipopolysaccharide; hDPCs: human dental pulp cells; miR-146b NC: miRNA mimic Negative Control #1; miR-146b mimic: miRNA mimic for hsa-miR-146b-5p; RLU: relative light unit.

**Figure 4 ijms-24-07433-f004:**
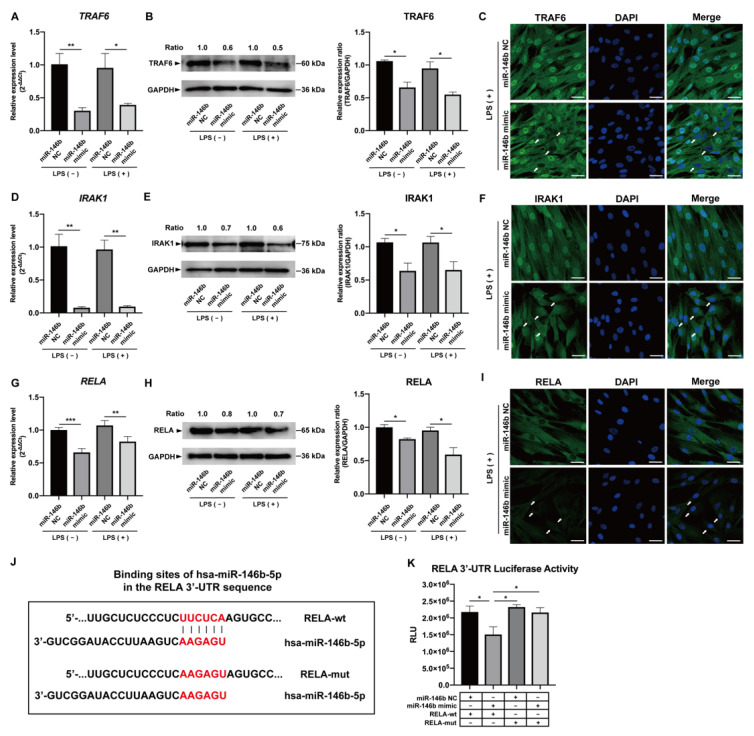
hsa-miR-146b-5p mimic down-regulates TRAF6, IRAK1, and RELA in LPS-stimulated hDPCs. hsa-miR-146b-5p mimic significantly down-regulated the mRNA (**A**,**D**,**G**) and protein (**B**,**C**,**E**,**F**,**H**,**I**) expression of TRAF6, IRAK1, and RELA in hDPCs under 2 h LPS stimulation (mean ± SD, n ≥ 3). (**J**) Wild-type (RELA-wt) and mutated (RELA-mut) target sequences of hsa-miR-146b-5p within RELA 3′-UTR are illustrated. (**K**) Luciferase reporter assay shows down-regulation of luciferase activity of RELA 3′-UTR in wild-type cells with over-expression of hsa-miR-146b-5p (mean ± SD, n = 4). * *p* < 0.05, ** *p* < 0.01, and *** *p* < 0.001. LPS: lipopolysaccharide; hDPCs: human dental pulp cells; miR-146b NC: miRNA mimic Negative Control #1; miR-146b mimic: miRNA mimic for hsa-miR-146b-5p; RLU: relative light unit; White arrows: target gene expression of TRAF6, IRAK1 or RELA was down-regulated in cytosol or nucleus in the white arrow indicated cells; Scale bars: 50 μm.

**Figure 5 ijms-24-07433-f005:**
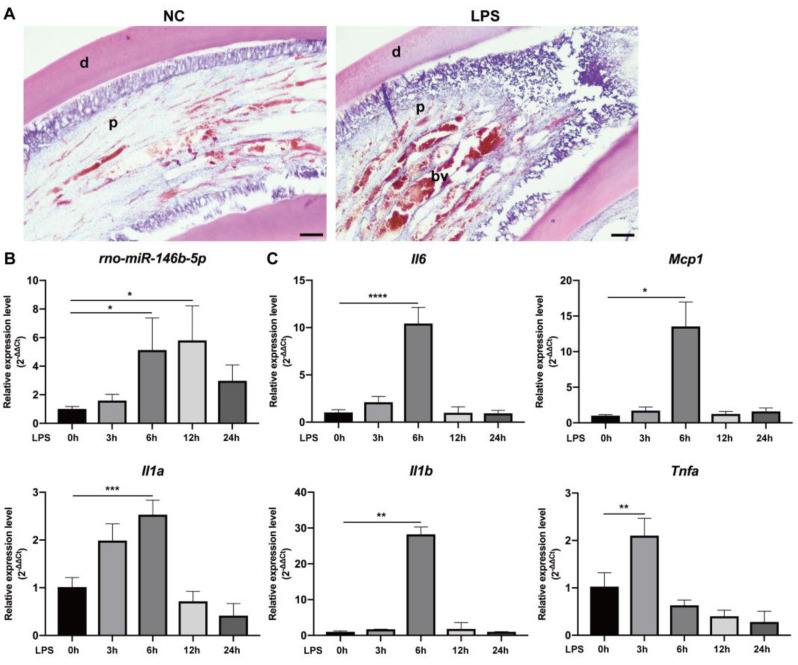
Expression of rno-miR-146b-5p and pro-inflammatory mediator mRNA is up-regulated in experimentally-induced rat pulpitis. (**A**) Typical inflammation reactions induced by LPS including vasodilatation and infiltration of inflammatory cells were observed in LPS-applied rat incisor pulp tissues (H&E). (**B**) Expression of rno-miR-146b-5p was significantly up-regulated in rat incisor pulp tissue at 6 h after LPS application and peaked at 12 h (mean ± SD, n = 3). (**C**) mRNA expression of pro-inflammatory mediators *Il6*, *Mcp1*, *Il1a*, *Il1b*, and *Tnfa* was induced by LPS stimulation and peaked at 6 h (*Tnfa* peaked at 3 h; mean ± SD, n = 3). * *p* < 0.05, ** *p* < 0.01, *** *p* < 0.001, and **** *p* < 0.0001. LPS: lipopolysaccharide; d: dentin; p: pulp; bv: blood vessel; Scale bars: 150 μm.

**Figure 6 ijms-24-07433-f006:**
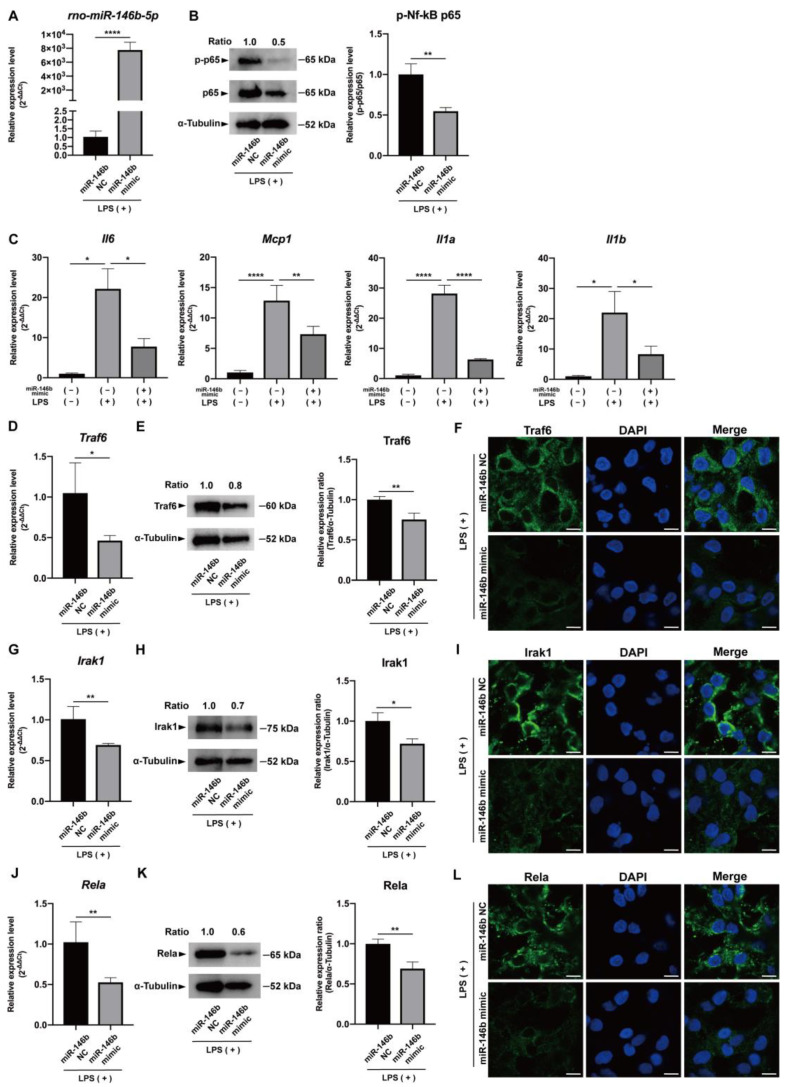
rno-miR-146b-5p mimic down-regulates mRNA expression of pro-inflammatory mediators in LPS-stimulated ex vivo cultured rat incisor pulp tissue. (**A**) Transfection of rno-miR-146b-5p mimic induced significant up-regulation of rno-miR-146b-5p in LPS-stimulated ex vivo cultured rat incisor pulp tissue (mean ± SD, n = 3). (**B**) rno-miR-146b-5p mimic significantly down-regulated the phosphorylation of NF-κB p65 in ex vivo cultured rat incisor pulp tissue with 6 h LPS stimulation (mean ± SD, n = 3). (**C**) rno-miR-146b-5p mimic significantly down-regulated the promoted mRNA expression of pro-inflammatory cytokines *Il6*, *Mcp1*, *Il1a*, and *Il1b* in ex vivo cultured rat incisor pulp tissue with 6 h LPS stimulation (mean ± SD, n = 3). (**D**,**G**,**J**) mRNA expression and (**E**,**F**,**H**,**I**,**K**,**L**) protein expression of Traf6, Irak1, and Rela were drastically down-regulated by over-expression of rno-miR-146b-5p in rat incisor pulp tissue 6 h after LPS stimulation (mean ± SD, n = 3). * *p* < 0.05, ** *p* < 0.01, and **** *p* < 0.0001. LPS: lipopolysaccharide; miR-146b NC: miRNA mimic Negative Control #1; miR-146b mimic: miRNA mimic for rno-miR-146b-5p; Scale bars: 5 μm.

**Figure 7 ijms-24-07433-f007:**
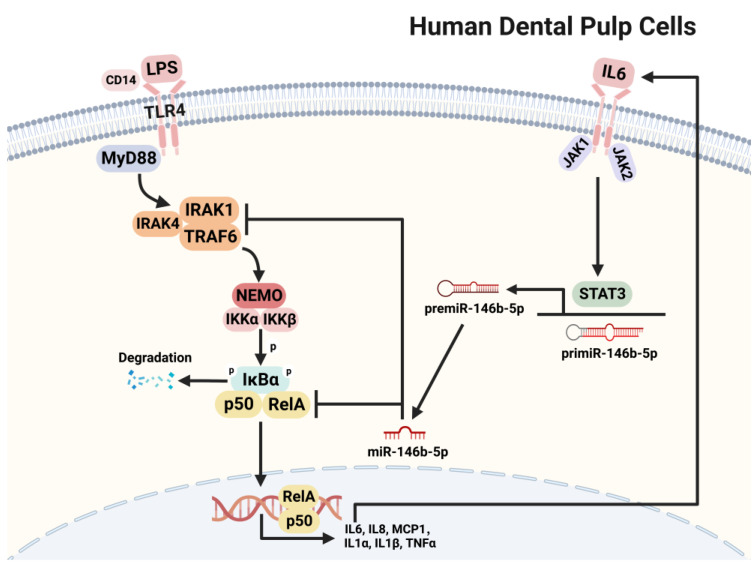
A schematic graph illustrating the role of miR-146b-5p in LPS-stimulated hDPCs. LPS binds to TLR4 thus activating the NF-κB signaling pathway and resulting in the synthesis of pro-inflammatory cytokines IL6, IL8, MCP1, IL1α, IL1β, and TNFα. Generally, IL6 further initiates JAK/STAT3 signaling, which is responsible to generate mature miR-146b-5p. miR-146b-5p, in turn, negatively regulates the NF-κB signaling pathway by targeting TRAF6, IRAK1, and RELA. The same phenomena were confirmed in LPS-applied rat dental pulp tissues ex vivo. Created with BioRender.com.

**Table 1 ijms-24-07433-t001:** Sequence for primers used in RT-qPCR.

Gene	Forward	Reverse	Accession No.	Size, bp
<human>				
*ACTB*	5′-GTAGCACAGCTTCTCCTTAATGTCA-3′	5′-CTGACTGACTACCTCATGAAGATCC-3′	NM_001101.3	102
*IL6*	5′-TATACCTCAAACTCCAAAAGACCAG-3′	5′-ACAAGAGTAACATGTGTGAAAGCAG-3′	NM_000600.4	157
*IL8*	5′-TCAGTGCATAAAGACATACTCCAAA-3′	5′-TCTTCCATCAGAAAGCTTTACAATAA-3′	NM_000584.4	121
*IL1A*	5′-GCAGTCACATACAATTGAGTTITTG-3′	5′-TTACATAATCTGGATGAAGCAGTGA-3′	NM_000575.4	116
*IL1B*	5′-CTCAAATTCCAGCTTGTTATTGATT-3′	5′-GTTGAAAGATGATAAGCCCACTCTA-3′	NM_000576.2	124
*MCP1*	5′-CACCTGCTGTTATAACTTCACCAAT-3′	5′-GTTGAAAGATGATAAGCCCACTCTA-3′	NM_002982.4	130
*TNFA*	5′-CCTGGTATGAGCCCATCTATCTG-3′	5′-GCAATGATCCCAAAGTAGACCTG-3′	NM_000594.3	130
*TRAF6*	5′-ACCCTCTAACTGGTGAATAGTTTCC-3′	5′-CTACAAGAGAACACCCAGTCACAC-3′	NM 145803.2	129
*IRAK1*	5′-ACAGAAGATGGTCCAGAAGCTG-3′	5′-AGCTCTGAAATTCATCACTTTCTTC-3′	NM_001569.3	140
*RELA*	5′-TTCCAAGTTCCTATAGAAGAGCAGC-3′	5′-TCAAAGATGGGATGAGAAAGGACAG-3′	NM_021975.4	134
<rat>				
*Actb*	5′-GTAAAGACCTCTATGCCAACACAGT-3′	5′-GGAGCAATGATCTTGATCTTCATGG -3′	NM_031144.3	127
*Il6*	5′-TAAGGACCAAGACCATCCAACTCAT-3′	5′-AGTGAGGAATGTCCACAAACTGATA-3′	NM_012589.2	125
*Il1a*	5′-CCAATCTGTACTGTTCACTTCGTTC-3′	5′-TTCCCGTCTTTAGATGGTTAGCTTT-3′	NM_017019.2	137
*Il1b*	5′-AGAAGAATCTAGTTGTCCGTGTGTA-3′	5′-GCTTGTGCTTCATTCATAAACACTC-3′	NM_031512.2	139
*Mcp1*	5′-CTAAGGACTTCAGCACCTTTGAATG-3′	5′-GTTCTCTGTCATACTGGTCACTTCT-3′	NM_031530.1	120
*Tnfa*	5′-AAACGGAGCTAAACTACCAGCTATC-3′	5′-CCTGGTCACCAAATCAGCATTATTA-3′	NM_012675.3	139
*Traf6*	5′-ATTGTTGAAATATGCTCTAGGCAGC-3′	5′-TGGAAACCAAGCTATACTGAACAGA-3′	NM_001107754.2	123
*Irak1*	5′-AAGTTCTCATGGTGTACAAACTCCT-3′	5′-GCCTTGTCTGTGCTTACATTATGAG-3′	NM_001127555.1	120
*Rela*	5′-CTTTCTCAAGTGCCTTAATAGCAGG-3′	5′-TTCAGAGCTAGAAAGAGCAAGAGTC-3′	NM_199267.2	121

Abbreviations: ACTB: actin beta; IL6: interleukin 6; IL8: interleukin 8; IL1A: interleukin 1 alpha; IL1B: interleukin 1 beta; MCP1: monocyte chemotactic protein 1; TNFA: tumor necrosis factor alpha; TRAF6: tumor necrosis factor receptor-associated factor 6; IRAK1: interleukin-1 receptor-associated kinase 1; RELA: REL-associated protein.

## Data Availability

The datasets that support the findings of the present study are available from the corresponding author upon reasonable request.

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
