# Peer review of "MicroRNA-146b-5p Suppresses Pro-Inflammatory Mediator Synthesis via Targeting TRAF6, IRAK1, and RELA in Lipopolysaccharide-Stimulated Human Dental Pulp Cells"

_ijms, 2023, doi:10.3390/ijms24087433_

Round 1
Reviewer 1 Report
In the paper entitled: "MicroRNA-146b-5p Suppresses Pro-inflammatory Mediator Synthesis via Targeting TRAF6, IRAK1, and RELA in Lipopolysaccharide-stimulated Human Dental Pulp Cells" the authors suggest that miR-146b-5p synthesis is controlled via a The IL6/STAT3 signaling cascade and, and in turn, miR-146b-5p down-regulates the expression of pro-inflammatory mediators by targeting TRAF6, IRAK1 and RELA in LPS-stimulated hDPCs.
The article is very interesting for the field of the topic. The manuscript is well-written and the topic is well-developed.
Minor
in sub-section 2.1 the authors should better clarify the correlation between the peak of miRNA overexpression and the peak of the cytokines RNA levels.
the authors should clarify how long the cells were stimulated with the LPS by section 2.2 and why different time points were used in the experiments.
Could the authors specify, in the text, the meaning of rno-miR-146b-5p ?
Major
In the schematic model represented in figure 7, the authors show a correlation between STAT3 and the process of mR-146b-5p maturation. In their work, they don't prove that STAT3 is really directly related to this process. On the base of their data, they speculate about this correlation also supported by literature data. Maybe it is true that this happens also in their experimental model, but they need to perform some experiments to support this hypothesis. So, I suggest carrying out the relative experiment to validate this interaction (i.e. RNA-immunoprecipitation or RNA pull down, to verify, at least, the interaction between miR-146b-5p and STAT3).
Reviewer 2 Report
Dear Authors,
This article revealed that the synthesis of miR-146b-5p is controlled via an IL6/STAT3 signaling cascade, and in turn, miR-146b-5p down-regulates the expression of pro-inflammatory mediators by targeting TRAF6, IRAK1, and RELA in LPS-stimulated hDPCs.
There are several questions below:
1. The authors already showed that miR-21-5p has anti-inflammatory effect for DPCs in the previous report. Is there any differences between miR-146b-5p and 21-5p about those anti-inflammatory effect and function?
2. In other cells and tissues except for dental pulp, there are a lot of reports related to miR-146a-5p and the mechanisms of anti-inflammatory effect. Did you try to examine miR-146a-5p in your experiments? Because miR-146a/b have strong anti-inflammatory effects in wide range of cells and tissues. That is to say, miR-146a/b is not so specific for anti-inflammatory effect.
3. Did you confirm IL-6 effect for promoting productions of pri-, pre- or miR-146b-5p in DPCs?
Sincerely yours,
Reviewer 3 Report
This manuscript study by Peifeng Han et al.
Present a “MicroRNA-146b-5p Suppresses Pro-inflammatory Mediator Synthesis via Targeting TRAF6, IRAK1, and RELA in Lipopoly-saccharide-stimulated Human Dental Pulp Cells”.
This manuscript well an organized story and a nice presentation of the data.
Authors show that miRNA-146b is involved NF-kB signaling pathway and regulates inflammation progress through the upstream IKK complex and downstream IKK complex.
Data presentation that TRAF6, IRAK2, and RELA are mainly regulated by regulation miR-146b.
Here's the thing, how about an IkB expression and phosphorylation also IKK alpha, beta, and gamma by miR-146b?
It will be strong evidence for the functional relevance of miR-146b.
Reviewer 4 Report
The study by Han et al. investigated the anti-inflammation effects of miR-146b-5p in lipopolysaccharide (LPS)-stimulated human dental pulp cells (hDPCs). The results showed that the expression of hsa-miR-146b-5p and pro-inflammatory cytokines was down-regulated by a nuclear factor-kappa B (NF-κB) inhibitor, and expression of hsa-miR-146b-5p was also decreased by a JAK1/2 inhibitor.
Following are some concerns which need to be addressed by the authors:
1. Include a better rationalization for choice of NF-κB inhibitor and JAK1/2 inhibitor.
2. Since there are several mechanisms are involved in pro-inflammatory cytokines, it is highly recommend adding some data about the effect of MAPK pathway.
3. Please check the typographical and grammatical errors. Ex. α-Tublin or α-Tubulin?
Reviewer 5 Report
Broad comments
The research question is interesting and the techniques used are widely accepted by the biomedical community and were well selected and performed by the authors. Authors performed and spend their time worthily, since both in human in vitro and in vivo animal model have been used. Molecular and protein targets were analyzed properly.
I appreciated the manuscript that can benefit of some review. Please, kind and expert authors, read and amend my following comments.
Major comments
1) Figure 3: panel B title has a typo, "avtivity"
2) TNFa in Figure 3 should be separated because it is not a gene expression result like the left inflammatory markers. Or that is an error on axis to be corrected?
3) figure 4 panel F (IRAK immunofluorescence) is not so clear to me, neither the label on the image. Can you better explain also in its figure legend what is the observation that means the result, i.e. the down-regulation?
4) titles similar to 2.4, 2.5, etc, should be more concise. Now they are not title, they are too long. Please, make shorter them.
5) Conclusion-related Figures (Biorender scheme) should include hDPCs in the cell cartoon and figure 7 legend (mention also rat tissues). The conclusion should be strictly contextualized. I guess the authors will approve also to confirm the novelty of their study and for auspicabile deserving publication.
Minor comments
6) What about a hypothetical priming strategy of the DP stem cells by using such investigated miR? can the authors discuss and suggest a perspective future investigation to change DP therapeutic effect , such as immunodulating and immunosuppressive MSC-like cells? Please consider to add reference for introducing the concept of priming of adult stem cells for regulate and modulate the immune response (10.3390/antibiotics10070750)
7)In both Figure 5 and Figure 6 cytokines are lowercase, but in other figures as uppercase. please correct and make homogeneous style.
8) in Figure 6 there is "Tublin", I am almost sure that it is typo to be changed in "tubulin". Right?
9) the authors can mention that many cells, for example also murine microglial cells (ref. doi.org/10.3390/antiox12040808, this can be added at line 214 together with reference n°14 ) , respond to TLR4 and changes it level upon LPS in vitro stimulation. They wrote at line 204 "selectively", but maybe it can be misinterpreted.
10) some limitations of this study can be listed honestly by the authors. no one work is perfect.
Round 2
Reviewer 2 Report
No comments
Reviewer 4 Report
I recommend to publish in this present form.
Reviewer 5 Report
All my comments were addressed after authors’ revision. The manuscript quality has been improved and I suggest as suitable to be published.